# The Importance of Forest Elephants for Vegetation Structure Modification and Its Influence on the Bird Community of a Mid-Elevation Forest on Mount Cameroon, West-Central Africa

Solange Mekuate Kamga [1,2,*], Simon Awafor Tamungang [1,3], Taku Awa II [1], Francis Luma Ewome [4], Francis Njie Motombi [5], David Hořák [6] and Jan Riegert [2]

1   Laboratory of Applied Biology and Ecology, Faculty of Science, University of Dschang, Dschang, West Region, Cameroon; atamungang@yahoo.com (S.A.T.); takuawa@yahoo.co.uk (T.A.II)
2   Department of Zoology, Faculty of Sciences, University of South Bohemia, Branišovská 1760, CZ-370 05 České Budějovice, Czech Republic; honza@riegert.cz
3   Department of Forestry and Wildlife Technology, College of Technology, The University of Bamenda, Bambili, Northwest Region, Cameroon
4   Bokwango, Buea, Southwest Region, Cameroon; lumaescobar@yahoo.com
5   Mount Cameroon National Park, Buea, Southwest Region, Cameroon; njiefrank@yahoo.com
6   Department of Ecology, Faculty of Science, Charles University, Viničná 7, CZ-128 44 Prague, Czech Republic; david.horak@natur.cuni.cz
*   Correspondence: kamgasolange1@gmail.com; Tel.: +237-699511912

**Abstract:** Most of the tropical rainforests are subject to both anthropogenic and natural disturbances. Forest elephants (*Loxodonta cyclotis*) cause forest clearings within the tropics. This study was conducted at mid-elevations (1100–1700 m a.s.l.) in Mount Cameroon National Park. We assessed the difference in the structure of bird communities in the forest and areas located nearby affected by elephant activities. We used the point-count method; 22 points were established within each habitat. The vegetation was visually estimated within a 50 m radius surrounding each counting point. In total, 1603 birds from 85 species were recorded. The vegetation cover from 3 to 5 m at points with elephant activity was significantly lower compared to points without elephant activity. Bird species richness was significantly higher around points in pasture compared to points in intact forest. Habitat type and the percentage of vegetation layer from 3 to 5 m significantly impacted the bird community structure. The points in the pasture were especially characterized by the increased abundance of some open habitat species (e.g., Chubb's *Cisticola chubbi*). Few studies have documented the effects of elephant activity on other species, although the effects are widely stated as important drivers of habitat diversity in tropical forests. In conclusion, disturbance caused by elephants leads to increased bird community diversity due to the increased heterogeneity of the environment, which documents the high importance of elephants as ecosystem engineers.

**Keywords:** diversity; bird community; forest elephants; vegetation structure; species composition

## 1. Introduction

Tropical rainforests, known to be among the most diverse ecosystem on Earth, function as the guarantors of life on earth through the many services they provide, such as climate regulation, maintenance of air quality, and biogeochemical cycles [1–3]. Although they cover less than 10% of the Earth's land area, tropical rainforests represent the largest reservoirs of biodiversity hosting diverse bird communities [2,4]. Unfortunately, fast human population growth and increasing demands for forest resources are putting immense pressure on rapidly vanishing rainforests [5–7]. The greatest disturbances are mostly caused by human activities [8], such as farming, bushfires, logging, and the introduction of invasive species [9]. These disturbances have a large impact on the structure of the forest,

and these changes represent new challenges for the survival of wildlife species, including birds [10].

The impact of anthropogenic activities on the diversity and composition of avifauna has been the center of attention over the past decades [11–16]. However, forests also face natural disturbances such as bark beetle outbreaks, forest fires, windstorms, and the impact of large animals [17–19]. Elephants are the largest terrestrial megafauna left on Earth [19,20]. They play an important role as ecosystem engineers, through the disruptive effects [19] of their foraging, altering the forest vegetation structure, which affects the temporal and spatial distribution of animals and plants [21]. Due to their large bodies and energy requirements, elephants are significant plant consumers [22,23] and influence the forest canopy cover by altering and reducing the density of plants [24]. These changes create new habitat types that serve as a niche for other organisms [25,26], with cascading effects on animal biodiversity [27], including avian communities [28]. Nevertheless, very few studies have examined the effect of elephants on the biodiversity of tropical forests showing the various effects of their activity on vertebrates, invertebrates, and soil properties in tropical forests [26,29].

Mount Cameroon represents a 58,178 ha block of the intact mountain environment and is one of the most diverse ecosystems in Cameroon. An active volcano, which has erupted six times [30] with the last eruption recorded in 2000 [31], means the soil is of recent origin and is very fertile with low water retention [31,32] It is also ranked among the 10 most protected and conserved areas throughout the world [33]. It hosts several endangered and endemic species of flora and fauna. Two bird species are endemic to the mountain—Mount Cameroon francolin *Pternistis camerunensis* and Mount Cameroon speirops *Zosterops melanocephalus* [34]. Of the approximately 137,000 forest elephants (*Loxodonta cyclotis*) in Central Africa [35], 176 individuals have been recorded on Mount Cameroon since 2003 [36]. Since then, several observations have been recorded yearly with increasing evidence of their impacts both within and outside the park well-documented but no update on the population status. Human disturbances are of minor importance there, because all the human activities such as tree logging or establishing new farms is forbidden within the area of Mount Cameroon National Park. As megaherbivores, elephants modify vegetation communities, changing species composition and plant cover [37] by uprooting, graying, and debarking feeding activities [38]. Some studies have suggested that elephant effects on large trees may reduce the nesting potential for tree-nesting bird species [39].

At the mid-elevation of the mountain, we selected sites with intact forest cover and those with pronounced indications of elephant activity. We hypothesized that the modification of vegetation structure by elephants affected the spatial distribution of birds due to changed quality of shelter against predators and food offer. Disturbed and more open forest understory vegetation has increased access to sunlight allowing it to thrive and grow. This in turn influences the distribution of food resources. We, therefore, expected that understory vegetation would attract a large proportion of forest bird species [40], which might even overbalance the canopy layer that has been previously reported to be a dominant center of bird abundance and diversity [41]. The principal aim of this study was to determine how changes in vegetation structure caused by elephants affect the forest bird community structure on Mount Cameroon.

## 2. Material and Methods

### 2.1. Study Area

Mount Cameroon represents the highest mountain in West and Central Africa. It forms a part of the Mount Cameroon National Park (MCNP) created in 2009, for the protection of endangered fauna and flora species from uncontrolled harvesting and poaching [31]. Our study was conducted in MCNP, South-West Region, Cameroon (N 4°9′36.5688″, E 9°16′44.9616″; 4095 m a.s.l.) at two locations known as Plantecamp and Crater Lake both located on the western slopes of the mountain. The area is characterized by a temperature above 18 °C [4,42]. The Mount Cameroon region experiences two climate types—a dry

season that lasts from November to May and a rainy season with heavy rains from June to October [43,44]. Along the elevation gradient, the mean temperature drops by about 0.6 °C for every 100 m of ascent. The humidity remains at 75–85% due to the influence of the sea, the presence of fog, and the formation of orographic clouds, which are very frequent in the area [38]. Given all these characteristics, it has been described by Payton [45], as one of the mountains in West Africa receiving the lowest amount of annual sunshine. Its southwestern part, Debundscha, is the second wettest place in the world receiving almost 10,000 mm of annual precipitation per year after Mawsynram in India [43,46]. The forest extends from the foot of the mountain to 2300 m a.s.l. (the tree line is relatively low due to recent volcanic activity) and is dominated largely by shrubs, trees, and grass.

### 2.2. Bird Diversity Survey

This study was carried out at elevations between 1100 m and 1700 m a.s.l. with different levels of forest elephant activity, during the rainy season (17–25 July 2021) We used a point-count method [47], to study the diversity of bird species in areas affected by elephant activity. Although commonly used in the field to sample bird populations, this method also gives good estimates of species abundances for analyses of habitat preferences [48]. The point-count method consisted of moving from one point to the next, stopping at the pre-established location and recording all birds seen and/or heard for 15 min [47–49]. We set 44 sampling points (Figure 1); the points were situated at least 150 m apart from each other to ensure the independence of observations. Thus, 22 points were randomly located in areas modified by elephants (pastures) and the other 22 in areas with little or no elephant impact (forest). In the latter areas, elephants moved through forest sites, but elephant activity did not change the biotope in these areas.

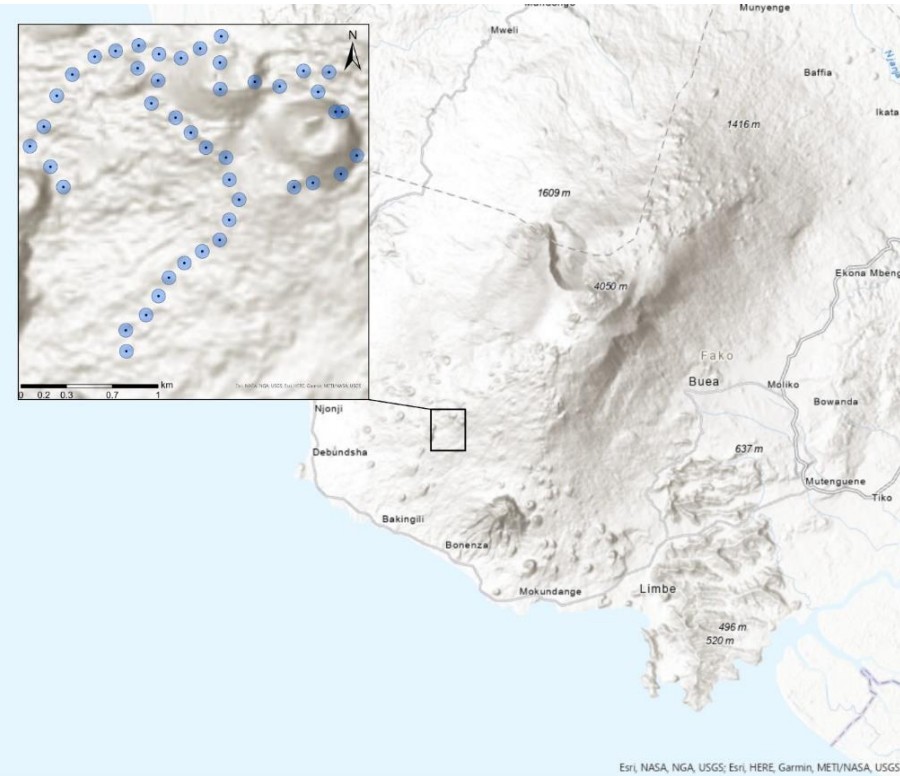

**Figure 1.** Map of the study area with positions of bird count points (ESRI hillshade basemap was used to create the figure).

We carried out two repeated surveys at each point after a minimum of two days to improve the quality of the community structure estimate. At each count point, birds were recorded within a radius of 50 m between 06:40 and 14:00, dependent on weather

conditions. The following parameters were also recorded at each count point: (i) species name, (ii) weather conditions, (iii) type of habitat, either pasture or forest, (iv) distance between the observers and the individual bird, (v) number of individuals seen or heard, and (vi) start and end times. A GPS was used to measure the distance between each count point and to record the coordinates for each observation point.

### 2.2.1. Vegetation Sampling

We carried out a visual estimation of vegetation cover along with the vertical forest profile within a 50 m radius around each sampling point. The vegetation cover was estimated for each stratum (0–1 m, 1–3 m, 3–5 m, 5–10 m, and above 10 m). For each observed cover layer we noted a percentage, representing the density of the vegetation in the stratum. We also included an estimate of the percentage of bushes at each of the 44 points.

### 2.2.2. Statistical Analyses

For basic statistical analyses, we used Statistica 13 Software, TIBCO, Palo Alto, CA, USA [50]. Differences in the cover of each vegetation layer between forest and pasture count points were computed using Mann-Whitney U tests. To determine correlations among the cover of each vegetation layer, we used Spearman's rank correlations. The purpose of this analysis was to uncover the correlation relationship between the targeted layer 3–5 m (i.e., the most affected layer by elephant activity) and other vegetation layers.

The effect of foggy weather on the detectability of individual birds was computed by GLM analyses for each species with the binomial dependent variable (i.e., presence/absence of each species) and independent variable "seen or heard". A data unit was represented by each count point for each counting period separately ($n = 88$ rows). Firstly, we built null models (i.e., without the independent variable). Then, we added the independent variable and compared it with the null models using an ANOVA function.

For further analyses on the bird community, we used the maximal abundance of each species from two consecutive counts at each count point, species with an abundance $\leq 3$ were considered insufficient for further analyses (excluding analyses on bird diversity). Before the analyses on bird community differences between forest and pasture count points, we used a distance sampling method to correct the abundances of all species according to their detectability based on the distance from the observer. These data were used to fit detection functions dfuncEstim and abundEstim using the package RDistance in R 4.0.3 software [51], following Macdonald et al. [52].

Alpha and beta diversity for forest and pasture count points was compared using the species richness, Simpson index [53], and Jaccard similarity [54] indices to compare the means and standard deviations between the two habitat types.

We used the Shannon index [55], to compare avian assemblage diversity according to habitat, with a GLM (R package "base") in the R software R Core Team with the dependent variable as bird diversity at each count point and the independent variable being forest and pasture habitat. A data unit was represented by each count point ($n = 44$). Firstly, we built a null model (i.e., without the independent variable). Then, we added the independent variable and compared both models using an ANOVA function.

Differences in the composition of the bird community between forest and pasture count points and cover of vegetation layers (i.e., primary predictors) were calculated using variance partitioning by principal coordinate analysis of neighbor matrices (PCNM) in Canoco 5 software [56], which was recommended by Marrot et al. [57]. This multivariate analysis enabled us to remove the effect of geographical position (i.e., space predictors) from the effect of primary predictors [58]. The analysis is suitable for calculating inter-correlated variables, since all these variables entered the analysis simultaneously. The analysis included nine steps: (1) primary predictor test (i.e., preliminary test of the overall effect of primary predictors on the dataset), (2) primary predictor testing by partial redundancy analysis (RDA) based on partial Monte-Carlo permutation tests ($n = 499$ per-

mutations), (3) principal coordinate analysis (PCoA) based on Euclidean distances (i.e., finding the main space predictors based on GPS coordinates), (4) PCNM for all predictors (i.e., preliminary test of the overall effect of space predictors on the dataset), (5) PCNM selection (i.e., the choice of space predictors based on coordinates using forward selection and partial Monte-Carlo permutation tests), (6) spatial effects analysis (i.e., assessing the amount of variability explained by space predictors), (7) primary predictor effects analysis (i.e., assessing the amount of variability explained by primary predictors), (8) joint effects analysis (i.e., assessing the amount of variability explained by both predictor types) and, (9) removal of spatial effects [59]. We included cover of each vegetation layer (0–1 m, 1–3 m, 3–5 m, 5–10 m, and above 10 m) and percentages of bushes within a buffer of 50 m radius around each count point as independent variables (i.e., predictors) in the PCNM analysis, a data unit was each count point ($n$ = 44). Differences in chosen species abundances from the PCNM graph that fitted the ordination axes at least by 4% were then computed using Mann-Whitney U test. Only significant results are shown in graphs. Relationships between chosen species abundances and cover of the vegetation layer 3 to 5 m were computed using regressions. Both these types of analyses were performed using Statistica 13 software.

## 3. Results

For most vegetation strata, we found differences in the amount of cover between forest and pasture count points, including the vegetation layer from 1 to 3 m and the percentages of bushes (Table 1). According to the Spearman Rank Order Correlation test, the stratum ranging from 3 to 5 m (i.e., forest elephant height) of vegetation showed either positive or negative correlation with most other strata (Table 2), and there was a difference between pasture and forest count points (Figure 2a).

**Table 1.** Results of the comparison of cover for each vegetation stratum between forest and pasture points ($n$ = 44), Mann-Whitney U test. Significant differences are provided in bold.

| Variable | U | *p* |
|---|---|---|
| Layer 0–1 m | 21.5 | **0.001** |
| Layer 1–3 m | 233.5 | 0.851 |
| Layer 3–5 m | 62.5 | **0.001** |
| Layer 5–10 m | 63.5 | **0.001** |
| Layer >10 m | 48.5 | **0.001** |
| % of bushes | 194 | 0.264 |

**Table 2.** Spearman's rank correlation coefficients among different vegetation layers. Statistically significant relationships at $p < 0.05$ are in bold. See Table 1 for vegetation heights for each layer.

| Variables | Layer 2 | Layer 3 | Layer 4 | Layer 5 | % of Bushes |
|---|---|---|---|---|---|
| Layer 1 | −0.17 | **−0.56** | **−0.56** | **−0.60** | **0.33** |
| Layer 2 | | **0.44** | 0.18 | −0.03 | 0.13 |
| Layer 3 | | | **0.74** | **0.49** | −0.07 |
| Layer 4 | | | | **0.76** | −0.06 |
| Layer 5 | | | | | −0.17 |

We recorded a total of 1603 individual birds across 44 sampling points with 85 species recorded. Of these, there were 34 families, mostly dominated by Pycnonotidae (23.5%), Nectariniidae (17.6%), and Cisticolidae (14.7%) each; the most commonly recorded species were Yellow-billed turaco (*Tauraco macrorhynchus*) and Western Mountain greenbul (*Arizelocichla tephrolaema*) each with 44 records in total. Singletons were recorded for 18 species (21.2% of all sampled species). Two endemic bird species were registered during the survey, Yellow-breasted boubou (*Laniarus atroflavus*) and the Mount Cameroon francolin (*Pternistis camerunensis*), which is endangered [33].

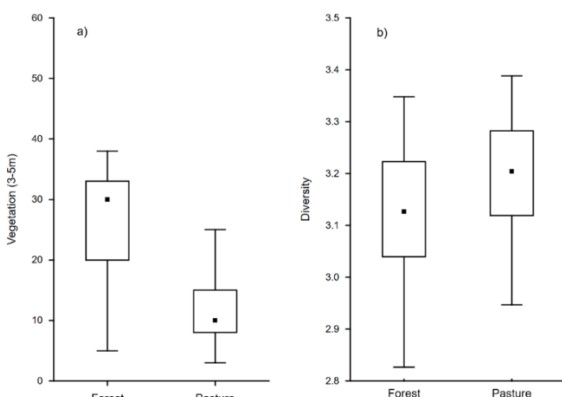

**Figure 2.** Differences in vegetation density (**a**) and species diversity (**b**) between both forest and pasture habitats. Points—medians, rectangles—25–75% of data, whiskers—non-outlier range.

The differences in bird abundances when comparing the field abundance versus corrected abundances by distance sampling are illustrated in Supplementary Table S1. For most of the species, the corrected abundances were higher than the observed ones. Those with a difference $\geq 7$ were Cameroon sunbird (*Cyanomitra oritis*), Western Mountain greenbul, Chubb's cisticola (*Cisticola chubbi*), Waller's starling (*Onychognathus walleri*), and African green pigeon (*Treron calvus*) (Supplementary Table S2).

The effect of fog on the detectability of birds (GLM analysis) was not significant for all bird species (Supplementary Table S2). Those with highly significant levels of explained variability were Naked face barbet (*Gymnobucco calvus*, d.f. = 13, 23.7% of explained variability, Chi = 3.55, $p = 0.005$), Grey chested illadopsis (*Kakamega poliothorax*, d.f. = 63, 15.8% of explained variability, Chi = 3.83, $p = 0.050$), Mountain robin chat (*Cossypha isabellae*, d.f. = 43, 6.8% of explained variability, Chi = 3.40, $p = 0.060$), Velvet mantled drongo, d.f. = 3.0% of explained variability, Chi = 2.91, $p = 0.081$), Yellow breasted boubou (*Laniarus atroflavus*, d.f. = 44, 15.05% of explained variability, Chi = 3.34, $p = 0.062$), and Grey headed greenbul (*Phyllastrephus poliocephalus*, d.f. = 11, 24.6% of explained variability, Chi = 3.46, $p = 0.064$). In most other species, their detection was not significantly influenced by the presence of fog (Supplementary Table S3).

Bird diversity significantly differed between forest and pasture points (GLM analysis, d.f. = 42, 8.6% of explained variability, Chi = 0.07, $p = 0.046$). Pasture points showed higher bird diversity compared to the forest points (Figure 2b). Using the alpha and beta diversity indices, the species richness was found to be higher in the pasture compared to the forest (Table 3). The tests of the Jaccard indices showed similar values for pasture and forest points.

**Table 3.** Alpha and beta diversity between forest and pasture points in terms of the species richness, Simpson index, and Jaccard similarity index.

|  | Forest | Pasture |
|---|---|---|
| Species richness | 22.64 ± 3.87 (15–29) | 27.00 ± 3.16 (21–32) |
| Simpson index | 0.07 ± 0.01 (0.05–0.10) | 0.07 ± 0.01 (0.05–0.11) |
| Jaccard index | 0.43 ± 0.11 (0.24–0.83) | 0.54 ± 0.08 (0.31–0.77) |

Using PCNM analysis, we found that the cover of the vegetation layer from 3 to 5 m (5.9% of explained variability, pseudo-F = 2.6, $p = 0.006$) and habitat type (3.6% of explained variability, pseudo-F = 1.6, F = 0.030) significantly affected the bird community (Table 4). The cover of the vegetation layer from 3 to 5 m was negatively correlated with the abundance of both Northern double collared sunbird (*Cinnyris reichenowi*, ß = −0.30, $R^2 = 0.08$, F = 3.12, $p = 0.080$) and Chubb's cisticola (ß = −0.42, $R^2 = 0.17$, F = 7.10, $p = 0.010$. Increased cover of the vegetation layer from 3 to 5 m also tended to attract Yellow-billed

barbet (*Trachylaemus purpuratus*), Tambourine dove (*Turtur tympanistria*), Black winged oriole (*Oriolus nigripennis*), Yellow whiskered greenbul (*Eurillas latirostris*), Yellow billed tauraco (*Tauraco macrorhynchus*), Velvet mantled drongo (*Dicrurus modestus*), and Yellow white eye (*Zosterops senegalensis*). Conversely, increased cover of the vegetation layer from 3 to 5 m had a negative effect on the abundance of Mountain sooty boubou (*Laniarus poensis*), Grey cuckooshrike (*Coracina caesia*), Common wattle eye (*Platysteira cyanea*), and Mountain saw-wing (*Psalidoprocne fuliginosa*) (Figure 3).

**Table 4.** The results of the PCNM analysis on the effect of environmental variables on the bird community structure. I and II ordination axes together explained 16.5% of variability. PCO—space predictor.

|  | Contribution % | Pseudo-F | *p* |
| --- | --- | --- | --- |
| Vegetation layer (3–5 m) | 12.3 | 2.6 | 0.006 |
| Habitat | 7.5 | 1.6 | 0.036 |
| PCO.3 | 10.7 | 2.9 | 0.006 |
| PCO.4 | 8.8 | 2.5 | 0.002 |
| PCO.7 | 7.1 | 2 | 0.008 |
| PCO.6 | 6.3 | 1.8 | 0.016 |

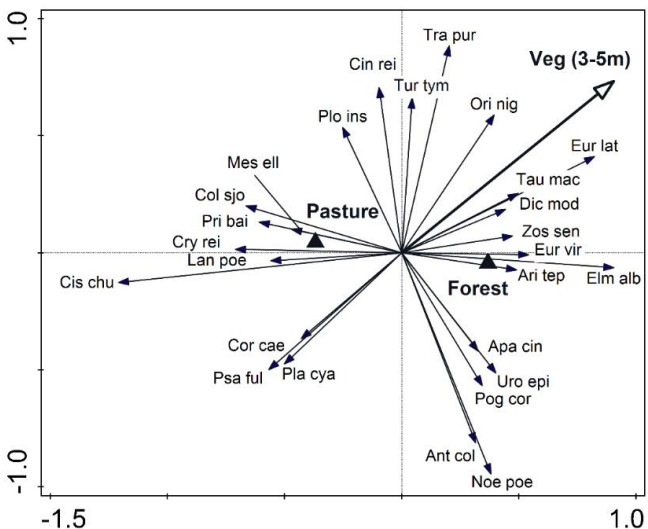

**Figure 3.** PCNM analysis visualizing the distribution of bird species (black-tipped arrows), habitat (triangles), and the vegetation (3-5 m, white-tipped arrow) in the space of the two axes indicating the valuable habitat type within a bird community in the tropical rainforest of Mount Cameroon. Species codes: Pla cya—*Platysteira cyanea*, Psa ful—*Psalidoprocne fuliginosa*, Cor cae—*Coracina caesia*, Cis chu—*Cisticola chubbi*, Lan poe—*Laniarus poensis*, Cry rei—*Cryptospiza reichnovii*, Pri bai—*Prinia bairdii*, Col sjo—*Columba sjostedti*, Mes ell– *Mesopicos elliotii*, Plo ins—*Ploceus insignis*, Cin rei—*Cinnyris reichenowi*, Tra pur—*Trachylaemus purpuratus*, Tur tym—*Turtur tympanistria*, Ori nig—*Oriolus nigripennis*, Eur lat—*Eurillas latirostris*, Tau mac—*Tauraco macrorhynchus*, Dic mod—*Dicrurus modestus*, Zos sen—*Zosterops senegalensis*, Eur vir—*Eurillas virens*, Elm alb—*Elminia albiventris*, Ari tep—*Arizelocichla tephrolaema*, Apa cin—*Apalis cinerea*, Uro epi—*Urolais epichlorus*, Pog cor—*Pogoniulus coryphaea*, Neo poe—*Neocossyphus poensis*, Ant col—*Anthodiaeta collaris*.

Abundances of Chubb's cisticola, Mountain sooty boubou, Waller's starling, and African hill babbler (*Pseudoalcipe abyssinica*) were significantly higher in the pasture points compared to the forest points (Table 5 and Figure 4). Moreover, other species such as Red-face crimsonwing, Banded prinia (*Prinia bairdii*), Cameroon olive pigeon (*Columba sjostedti*), Elliot's woodpecker (*Mesopicos elliotii*), Brown-capped weaver (*Ploceus insignis*), and Northern double collared sunbird, showed a strong association with pasture. On

the other hand, Little greenbul (*Eurillas virens*), White-bellied crested flycatcher (*Elminia albiventris*), Western Mountain greenbul, Grey apalis (*Apalis cinerea*), Green longtail (*Urolais epichlorus*), Western green tinkerbird (*Pogoniulus coryphaea*), Collared sunbird (*Anthodiaeta collaris*), and White-tailed ant thrush (*Neocossyphus poensis*) were mostly seen or heard at the forest points.

**Table 5.** Comparison of abundances of the most common species between forest and pasture points (*n* = 44). Mann Whitney U tests.

| Bird Species | U | *p* |
|---|---|---|
| *Cisticola chubbi* | 56.5 | 0.003 |
| *Laniarus poensis* | 36 | 0.013 |
| *Onychognathus walleri* | 55 | 0.017 |
| *Pseudoalcippe abyssinica* | 123.5 | 0.044 |

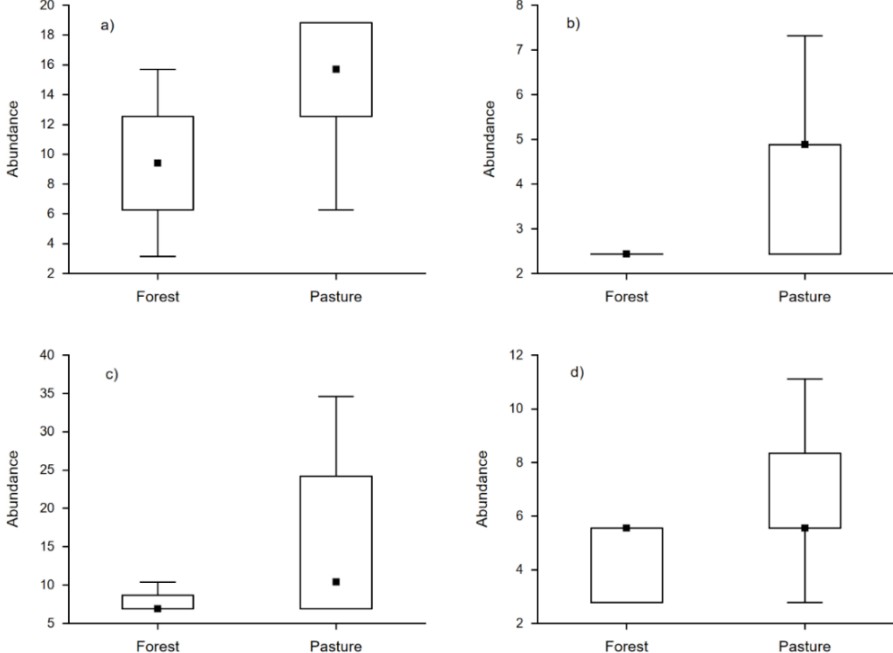

**Figure 4.** Differences in abundances between forest and pasture for species with increased abundances in pastures: (**a**) Chubb's cisticola; (**b**) Mountain sooty boubou; (**c**) Waller's starling; (**d**) African hill babbler. Points—medians, rectangles—25–75% of data, whiskers—non-outlier range.

## 4. Discussion

We show that the structure of avian assemblages significantly differed between forest and elephant pasture count points. This was mostly due to species richness, which was higher in the pasture points compared to the forest points. We found no important differences for avian beta diversity between the biotopes. Finally, although pastures were probably created relatively recently and were surrounded by dense tropical forest biotope, some species found them suitable habitat and showed apparently higher abundances there than in the pristine forest.

We also tested the effect of fog on the detectability of bird species, which interestingly was significant only for Chubb's cisticola, which was more active during the foggy weather. This result can be interpreted as (1) the probability for a predator detecting the bird in the fog is very low or (2) because of their shy behavior they prefer to carry out their activities when the climate is not favorable (e.g., due to wetting their feathers). Unfortunately, no study has yet been performed on the behavior of birds in foggy climates.

We did not find differences in beta diversity between the pasture and the forest points. This indicates that spatial turnover of avian assemblages does not change after elephants

"cut the forest down". Of course, the human perspective of the situation might be different from a bird's perspective, and there are also differences among species. According to Okland [60], not all birds perceive changes in their habitat in the same way, for some it increases the availability of food resources, and for others it hinders their survival. The fragmentation of forest habitats consequently leads to local species extinctions since it reduces forest cover [61]. There are also generalist species that do not show any preference for one habitat [62]. In our study, these species were represented by Tambourine dove, Black-winged oriole, and Yellow-billed barbet.

Obviously, the elephant pastures and forests differ in vegetation structure. We noticed that the cover of young trees (i.e., the vegetation layer from 3 to 5 m) was strongly reduced in the pasture sites compared to the forest sites. According to our observations, the pasture points were dominated by plants of the genus *Afromomum* (Zingiberaceae). Our results suggest that the elephants' pastures are not completely unsuitable for Mount Cameroon forest birds, as the environmental changes caused by elephants were not so profound as to cause the loss of a bird community, as was found by other studies, e.g., [24]. Moreover, Herremans [29], supported the evidence that more species of birds inhabited open landscape as opposed to less affected areas. Although increased vegetation heterogeneity or floristic diversity of a given habitat does not always lead to increased species diversity [63,64], the vast majority of studies show that heterogeneity of vegetation supports increased diversity of animals [65,66]. In birds, Greenberg and Lanham [67] demonstrated that forest openings created a suitable space for various species. Moreover, Bazzaz [68] stated that vegetation with a complex structure increases the diversity of food allowing the exploitation of diverse environmental resources. Thus, the availability of diverse microenvironments and/or food resources within the pastures potentially gives rise to increased bird species diversity [24]. Bird species, such as the Yellow-billed turaco and Velvet mantled drongo (*Dicrurus modestus*), were often found in the 3 to 5 m vegetation layer. Additionally, the understory layer of the disturbed vegetation has increased access to sunlight allowing it to thrive and grow, attracting more insects [69]. This is consistent with our result, since most of the bird species recorded in the pasture points were either insectivores (e.g., Brown-capped weaver and Yellow-breasted boubou) or habitat specialists (e.g., Chubb's cisticola and Red-faced crimsonwing). On the contrary, Hassan et al. [70] pointed out that human disturbances lead to a very important loss of vegetation and, thus, negatively influence the bird community. However, we still lack comparison of the effects of human and elephant disturbances on forest bird communities.

Some species had higher population densities in the pastures than in the forest. This is interesting especially because the forest is the original habitat for almost all species recorded; thus, we assumed they would thrive better there. Within the pastures, we recorded the highest densities of Chubb's cisticola, Mountain Sooty boubou, Waller's starling, and African hill babbler (*Pseudoalcippe abyssinica*). The high abundance of Chubb's cisticola is very likely the result of its habitat preferences; the species is in fact an open to shrub land specialist across the Cameroonian mountains with a tolerance for disturbed habitats [71,72]. On the other hand, Mountain Sooty boubou and African hill babbler are forest specialists, for which high densities outside the true forest might be surprising. However, both of these are in fact confined to the bushes in the forest undergrowth [73]. Thus, opening of the forest, which allows sun to reach lower vegetation strata, might even support the food availability of these birds and make their population more viable. Similarly, a forest specialist Yellow breasted boubou (*Laniarius atroflavus*) was reported abundantly in a mosaic landscape of the Bamenda Highlands [74]. The increased abundance of Waller's starling could potentially be a result of higher visibility in the pastures. It moves in flocks in the forest canopy [73], therefore, its presence may be affected by the random occurrence of a larger flock or better visual accessibility to the canopy layer in open spaces.

Our conclusions also corroborate previous work by MacArthur and MacArthur [75], who found that when more strata are present, higher diversity can be expected. Although it is difficult to find an open landscape bird community in the study area, as the vast majority

of birds are forest specialists, the effect of the relationship between two or more habitats, the so-called "edge effect" [76], can theoretically explain much of the increased diversity in pasture points. In fact, this effect creates an ecological corridor that allows species to move from one habitat type to another. Moreover, other studies [77–79] showed that forest edges positively affected the livelihoods of birds, which increased both their abundances and the species richness. In contrary, Robinson [80] suggested that forest edge is a threat to breeding birds because of increased parasitism and nest predation rates. For example, Batary and Baldi [80], found that the degree of nest predation was lower within forest patches compared to the forest edges. On Mount Cameroon, predation rates of avian nests seem to decrease with elevation [81], with no signature increase at mid-elevations, where the forest pastures occur.

In conclusion, we found that the moderate natural disturbances caused by forest elephants within the forest on Mt. Cameroon increased bird diversity as a result of a mixture of open and closed habitats. This can contribute to higher population densities in some species, even though some true forest specialists suffer due to forest area reduction and changes in vegetation structure. Findings from this study will draw attention to the important role of elephants as a keystone species and, hopefully, prompt park management to update its population status and distribution within the Mount Cameroon National Park.

**Supplementary Materials:** The following supporting information can be downloaded at: https://www.mdpi.com/article/10.3390/d14030227/s1, Table S1: Abundances of the species from field survey and after correction using distance sampling method; Table S2: Results of GLM analyses on the effect of fog (0/1) on the detectability of each species (0/1); Statistically significant ($p < 0.05$) and marginally significant ($p < 0.10$) are in bold. Table S3: Summary of species observations relative to fog presence.

**Author Contributions:** S.M.K. conceived, collected field data, participated in data analyses, and wrote the manuscript. S.A.T. supervised the research, improved the study design, edited, and revised the final draft. J.R. co-supervised the research, co-designed the study, analyzed the data, and participated in writing the manuscript. T.A.II co-designed the study, edited, and revised the final draft. D.H. co-designed the study, financially supported the fieldwork, edited, and revised the final draft. F.L.E. and F.N.M. collected field data. J.R. prepared the graphical abstract with recommendations from S.M.K. All authors have read and agreed to the published version of the manuscript.

**Funding:** This research was funded by the Czech Science Foundation (GACR 21-17125S).

**Institutional Review Board Statement:** Ethical review and approval were waived for this study, because birds were observed without disruption of their integrity.

**Informed Consent Statement:** Not applicable.

**Data Availability Statement:** Data are available in Supplementary Materials.

**Acknowledgments:** We also thank Ingrid Steenbergen for the language corrections.

**Conflicts of Interest:** The authors declare no conflict of interest.

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
