# Peer review of "The Importance of Forest Elephants for Vegetation Structure Modification and Its Influence on the Bird Community of a Mid-Elevation Forest on Mount Cameroon, West-Central Africa"

_diversity, doi:10.3390/d14030227_

Round 1
Reviewer 1 Report
This is a very interesting study on a rather poorly covered subject in the literature; therefore it is an addition certainly worth publishing.
I do not know the habitats of Mount Cameroon NP and therefore my ignorance makes that I would prefer to see some more explanations in the text (which are easy to make).
The abstract mentions "forest fragmentation" (line 16) caused by elephants. In my understanding the term fragmentation is mostly used in a context where the intervening hostile environment causes a barrier to animal/plant movement/dispersal. The fact that beta diversity is the same in both habitats indicates that the pastures do not function as barriers; so maybe thet term fragmentation is not appropriate here. Maybe forest clearings/openings are better? Adding a clearer description (or even better, typical pictures in supplemental material would help clarify what is exactly meant by "pastures" and how big they are (I guess from google images that they are clearings in the forest, mostly some 100-500 m across ?)).
The introduction starts with a paragraph of human disturbance. But little is done with that further on in the paper (not even in the discussion). It raises the question wether human disturbance of the forest and conversion to "pastures" might have a similar beneficial effect as natural disturbance by elephants (I guess certainly not!). Please add to discussion or consider removing human disturbance from introduction (as we are not talking about the same thing most likely).
The title and the article focusses on "elephant impact on forest and biodiversity". But it is insufficienty made clear to what extent elephants are solely responsible for the creation and/or maintenance of the forest pastures. What is the effect of soil and/or other animals, windfall, ... ?
The surveys took place in the wet season. Congratulations with that: It takes some perseverance to work in mountain cloud forest in the wet season! But is there a good reason to do so? Did the surveys coincide with the breeding season of most species (or is breeding not particularly seasonal in this habitat)? and therefore did it coincide with the peak of vocal activity and territoriality of most species? This also raises the question wether most of the records relate to resident birds in their breeding habitat. Were mixed bird flocks moving around also included? Or do they not exist at this time of year? This remark is related to the point in the discussion about the flock of Waller's starling. Flocks of non-breeding birds are a nuisance to habitat comparisons.
I could also do with some clarification of the field methods. In my experience distance sampling is extremely difficult in closed (tropical) forest, except if you use a small count cirkel (e.g. only 10-25 m and disregard everything beyond). In the open grasslands visibility and detectability is inevitably much better. How did you work out the recording distances? In well structured habitat with good visibility and excellent digitized aerial photo's, it is very easy to do, but in tropical forest?
Up to what distances were birds recorded? and what did you do when birds were detected from the forest (in the distance), while counting from a point in the pasture (or the reverse)? If those are systematically all included as part of that count point, regardless in what habitat they were located in, it all becomes a matter of scale. If clearings indeed have some additional birds associated with that habitat or the ecotone effect, then it is obvious that diversity is greater at these points if you add them up with the forest birds you also record around the clearing, and compare that to places with only forest.
Birds with less than 4 individuals were disregarded. Are the also excluded from the total figures given? How many species were in that group?
You mention endemics briefly. Is it not possible to identify a broader suit of species that are endemic/near endemic/regional characteristic (or some other sort of measure that makes them more important to local conservation) and evaluate the effects of pastures on them?
Line 379-384: the fog discussion is brief, but interesting. In the temperate world, woodland birds keep remarkably quite during thick fog. Possibly because it makes them wet and heavy and hampers swift movement, but probably also because this handicap and the fact that they cannot detect avian predators timely make them more vulnerable and hence more prudent.
some minor comments on the text:
line 193: is it not better to say "the most commonly recorded species", because I guess this is before distance correction?
line 257 Figure 3 Pog cor needs a capital in the figure
Author Response
Reviewer 1
This is a very interesting study on a rather poorly covered subject in the literature; therefore it is an addition certainly worth publishing.
Authors: Thank you
I do not know the habitats of Mount Cameroon NP and therefore my ignorance makes that I would prefer to see some more explanations in the text (which are easy to make).
Authors: We have added more details on the L78-79, 81, 114-116.
The abstract mentions "forest fragmentation" (line 16) caused by elephants. In my understanding the term fragmentation is mostly used in a context where the intervening hostile environment causes a barrier to animal/plant movement/dispersal. The fact that beta diversity is the same in both habitats indicates that the pastures do not function as barriers; so maybe thet term fragmentation is not appropriate here. Maybe forest clearings/openings are better? Adding a clearer description (or even better, typical pictures in supplemental material would help clarify what is exactly meant by "pastures" and how big they are (I guess from google images that they are clearings in the forest, mostly some 100-500 m across ?)).
Authors: Done, we mentioned forest clearings.
The introduction starts with a paragraph of human disturbance. But little is done with that further on in the paper (not even in the discussion). It raises the question wether human disturbance of the forest and conversion to "pastures" might have a similar beneficial effect as natural disturbance by elephants (I guess certainly not!). Please add to discussion or consider removing human disturbance from introduction (as we are not talking about the same thing most likely).
Authors: Done. We have included this aspect in the discussion (L435-438).
The title and the article focusses on "elephant impact on forest and biodiversity". But it is insufficienty made clear to what extent elephants are solely responsible for the creation and/or maintenance of the forest pastures. What is the effect of soil and/or other animals, windfall, ... ?
Authors: We added sentence into discussion
The surveys took place in the wet season. Congratulations with that: It takes some perseverance to work in mountain cloud forest in the wet season! But is there a good reason to do so? Did the surveys coincide with the breeding season of most species (or is breeding not particularly seasonal in this habitat)? and therefore did it coincide with the peak of vocal activity and territoriality of most species? This also raises the question wether most of the records relate to resident birds in their breeding habitat. Were mixed bird flocks moving around also included? Or do they not exist at this time of year? This remark is related to the point in the discussion about the flock of Waller's starling. Flocks of non-breeding birds are a nuisance to habitat comparisons.
Authors: The study of Chmel et al (2021) on mount Cameroon documents that the diversity of bird species in the dry season is the same as in the rainy season. Mixed flocks were not recorded, we rarely recorded one species flocks (e.g. Waller's starling).
I could also do with some clarification of the field methods. In my experience distance sampling is extremely difficult in closed (tropical) forest, except if you use a small count cirkel (e.g. only 10-25 m and disregard everything beyond). In the open grasslands visibility and detectability is inevitably much better. How did you work out the recording distances? In well structured habitat with good visibility and excellent digitized aerial photo's, it is very easy to do, but in tropical forest?
Authors: Most of individuals were detected by voice (80%) and the rest were seen.
Up to what distances were birds recorded? and what did you do when birds were detected from the forest (in the distance), while counting from a point in the pasture (or the reverse)? If those are systematically all included as part of that count point, regardless in what habitat they were located in, it all becomes a matter of scale. If clearings indeed have some additional birds associated with that habitat or the ecotone effect, then it is obvious that diversity is greater at these points if you add them up with the forest birds you also record around the clearing, and compare that to places with only forest.
Authors: Birds were sampled within a 50m radius, the distance was measured with a laser range finder without problem also in dense forest.
Birds with less than 4 individuals were disregarded. Are the also excluded from the total figures given? How many species were in that group?
Authors: These species were included only into Fig. 2, where we show bird community diversity.
You mention endemics briefly. Is it not possible to identify a broader suit of species that are endemic/near endemic/regional characteristic (or some other sort of measure that makes them more important to local conservation) and evaluate the effects of pastures on them?
Authors: Only two endemic bird species exist on mount Cameroon. Apart from those two that were recorded during the survey, the others were Least Concern.
Line 379-384: the fog discussion is brief, but interesting. In the temperate world, woodland birds keep remarkably quite during thick fog. Possibly because it makes them wet and heavy and hampers swift movement, but probably also because this handicap and the fact that they cannot detect avian predators timely make them more vulnerable and hence more prudent.
Authors: Done, a sentence has been added on L399.
some minor comments on the text:
line 193: is it not better to say "the most commonly recorded species", because I guess this is before distance correction?
Authors: Done.
line 257 Figure 3 Pog cor needs a capital in the figure
Authors: Done.
Reviewer 2 Report
Thanks for the opportunity to review such an interesting paper. As an elephant specialist I'm pleased to see this quantification of elephant roles in community diversity. I had some minor suggestions on the structure for readability in the intro (see sticky notes on PDF), and some general comments too, but I do think your stats methods and results sections needs more work before this paper is ready for publication. With the changes I've suggested, I think this is going to be of interest to a broad readership, and I think you have some good results here.
Your abstract needs to mention that few studies have documented the effects of elephant activity on other species, although the effects are widely stated as important drivers of habitat diversity in tropical forests.
Ln 54-58 is a repeat of Ln 50-52, so I would put all that info together; this is an important paragraph because it sets the context of the study:
"However, forests also face natural disturbances such as bark beetle outbreaks, forest fires, windstorms and the impact of large animals [17-19]. Elephants are the biggest terrestrial megafauna left on Earth [19-20]. They play an important role as great ecosystem engineers, through disruptive effects [19] of their foraging, altering the forest vegetation structure, which affects the temporal and spatial distribution of animals and plants [21]. Due to their large bodies and energy requirements, elephants are significant consumers [22,23] and influence the forest canopy cover by altering and reducing the density of plants [24]. These changes create new habitat types that serve as a niche for other organisms [25,26], with cascading effects on animal biodiversity [27], including avian communities [28]. Nevertheless, very few studies have examined the effect of elephants on the biodiversity of tropical forests showing the various effects of their activity on vertebrates, invertebrates and soil properties in tropical forests [26,29]."
Your stats methods should broadly follow the order that your methods and results follow, and at the moment it's very confusing because there's such a lot of information listed. I'm not clear why so many analyses were done in some places (see sticky notes on PDF), but at the moment, it seems that the Vegetation cover analyses should come first, then the detectability analysis for weather effects, then the avian community results. At the moment, it feels like the data may have been over-analysed, and that's partly because the rationale aren't clear, and some basic information is missing e.g. why you chose the vegetation strata categories that were used (maybe just because those are standard, but for readers not used to vegetation analysis, that needs to be specified).
I'm not sure why you chose to analyse the alpha and beta diversity with 4 difference indices, and then a Shannon index by habitat type. These are effectively the same analyses 5 times over, to answer the question "is diversity different between forest and pasture habitats", and really what your paper is about is the corresponding avian diversity. I think you should revise the analyses to only include one or two measures that support your take-home - that elephant-created habitat patches increase avian assemblage diversity.
I'm not familiar with the PCA-style analysis you chose to analyse bird community composition, so some of my comments might reflect that, and just be a case of making information clearer to readers. I think you did a good job of explaining the rationale of choosing this analysis, and the steps involved; but I got confused about what was meant by "chosen species" (see sticky notes on manuscript) and by the vegetation strata identified. I think some of this would be helped by reorganising as I suggest above.
Overall the Discussion lacks impact and isn't well structured. Try and get rid of unnecessary joiners like "furthermore" and "also", especially at the start of sentences, and go back to what you were trying to show. You don't have to cite every study; some can just be in the Intro and maybe others can be left out. Keep it focused on what you found, how that fits to the current literature and especially the new contribution you made - in the Abstract and Intro you nicely set up that elephants are ecosystem engineers, but that the size of those effects are poorly documented, but this isn't a theme for your Discussion. Rather the Discussion seems to suggest elephants are bad for bird communities (which might be, but then should be at least mentioned in the intro), and then you get into a lot of confusing "this study said this, and this said that". I didn't feel there was a clear take home message for the reader. It may also be that reorganising the paper flow helps structure this better, as things get dealt with in a better order.
Bold significant P values in table S1.
I think you can collapse the fog-affected detection analyses in supplementary methods to a single table, so you can have 3 instead of 8 tables: and caption it:
Summary of bird species observations significantly affected by weather conditions (fog)
|
|
|
No fog |
Fog |
|
|
Velvet mantled drongo |
Dicrurus modestus |
Not seen |
1 |
2 |
|
Seen |
2 |
0 |
||
|
Naked face barbet |
Gymnobucco calvus |
Not seen |
10 |
3 |
|
Seen |
3 |
0 |
||
|
Grey chested illadopsis |
Kakamega poliothorax |
Not seen |
32 |
30 |
|
Seen |
3 |
0 |
||
|
Yellow breasted boubou |
|
|
|
|
|
|
|
|
|

Author Response
Reviewer 2
Thanks for the opportunity to review such an interesting paper. As an elephant specialist I'm pleased to see this quantification of elephant roles in community diversity. I had some minor suggestions on the structure for readability in the intro (see sticky notes on PDF), and some general comments too, but I do think your stats methods and results sections needs more work before this paper is ready for publication. With the changes I've suggested, I think this is going to be of interest to a broad readership, and I think you have some good results here.
Authors: Thank you.
Your abstract needs to mention that few studies have documented the effects of elephant activity on other species, although the effects are widely stated as important drivers of habitat diversity in tropical forests.
Authors: Done.
Ln 54-58 is a repeat of Ln 50-52, so I would put all that info together; this is an important paragraph because it sets the context of the study:
"However, forests also face natural disturbances such as bark beetle outbreaks, forest fires, windstorms and the impact of large animals [17-19]. Elephants are the biggest terrestrial megafauna left on Earth [19-20]. They play an important role as great ecosystem engineers, through disruptive effects [19] of their foraging, altering the forest vegetation structure, which affects the temporal and spatial distribution of animals and plants [21]. Due to their large bodies and energy requirements, elephants are significant consumers [22,23] and influence the forest canopy cover by altering and reducing the density of plants [24]. These changes create new habitat types that serve as a niche for other organisms [25,26], with cascading effects on animal biodiversity [27], including avian communities [28]. Nevertheless, very few studies have examined the effect of elephants on the biodiversity of tropical forests showing the various effects of their activity on vertebrates, invertebrates and soil properties in tropical forests [26,29]."
Authors: Done.
Your stats methods should broadly follow the order that your methods and results follow, and at the moment it's very confusing because there's such a lot of information listed. I'm not clear why so many analyses were done in some places (see sticky notes on PDF), but at the moment, it seems that the Vegetation cover analyses should come first, then the detectability analysis for weather effects, then the avian community results. At the moment, it feels like the data may have been over-analysed, and that's partly because the rationale aren't clear, and some basic information is missing e.g. why you chose the vegetation strata categories that were used (maybe just because those are standard, but for readers not used to vegetation analysis, that needs to be specified).
Authors: Done. We made clear all the issues.
I'm not sure why you chose to analyse the alpha and beta diversity with 4 difference indices, and then a Shannon index by habitat type. These are effectively the same analyses 5 times over, to answer the question "is diversity different between forest and pasture habitats", and really what your paper is about is the corresponding avian diversity. I think you should revise the analyses to only include one or two measures that support your take-home - that elephant-created habitat patches increase avian assemblage diversity.
Authors: Done, Sorensen and Renkonen indices were excluded.
I'm not familiar with the PCA-style analysis you chose to analyse bird community composition, so some of my comments might reflect that, and just be a case of making information clearer to readers. I think you did a good job of explaining the rationale of choosing this analysis, and the steps involved; but I got confused about what was meant by "chosen species" (see sticky notes on manuscript) and by the vegetation strata identified. I think some of this would be helped by reorganising as I suggest above.
Authors: Further explanation has been provided on L222-223.
Overall the Discussion lacks impact and isn't well structured. Try and get rid of unnecessary joiners like "furthermore" and "also", especially at the start of sentences, and go back to what you were trying to show. You don't have to cite every study; some can just be in the Intro and maybe others can be left out. Keep it focused on what you found, how that fits to the current literature and especially the new contribution you made - in the Abstract and Intro you nicely set up that elephants are ecosystem engineers, but that the size of those effects are poorly documented, but this isn't a theme for your Discussion. Rather the Discussion seems to suggest elephants are bad for bird communities (which might be, but then should be at least mentioned in the intro), and then you get into a lot of confusing "this study said this, and this said that". I didn't feel there was a clear take home message for the reader. It may also be that reorganising the paper flow helps structure this better, as things get dealt with in a better order.
Authors: We greatly rearranged our method, result and discussion according to recommendations and simultaneously we avoided the use of additional term to focus just on results and comparison with our study.
Bold significant P values in table S1.
Authors: Done.
I think you can collapse the fog-affected detection analyses in supplementary methods to a single table, so you can have 3 instead of 8 tables: and caption it:
Summary of bird species observations significantly affected by weather conditions (fog)
|
|
|
|
No fog |
Fog |
|
Velvet mantled drongo |
Dicrurus modestus |
Not seen |
1 |
2 |
|
Seen |
2 |
0 |
||
|
Naked face barbet |
Gymnobucco calvus |
Not seen |
10 |
3 |
|
Seen |
3 |
0 |
||
|
Grey chested illadopsis |
Kakamega poliothorax |
Not seen |
32 |
30 |
|
Seen |
3 |
0 |
||
|
Yellow breasted boubou |
|
|
|
|
|
|
Authors: Done.
Round 2
Reviewer 2 Report
Thanks for making the changes - I think it has really improved the flow of the paper. I had a couple of small suggestions again re flow of information - one was just that the edits had split up a sentence. Otherwise I think this is a nice paper, and I congratulate you all on producing such a nice study.

Author Response
Dear reviewer,
we accepted both the minor changes and we thank again for Your suggestions. Please see the improved version of the manuscript
best regards
Solange Kamga & Jan Riegert